



Natural Hazards
and Earth System
**New Approaches to Seismic Microzonation Modelling of Ground Shaking Using Direct**
**Characteristics of Influencing Criteria: Case Study of Bam City, Iran**
Reza Hassanzadeh[1], Mehdi Honarmand[2], Mahdieh Hossienjani Zadeh[3], Farzin Naseri[4]
Department of Ecology, Institute of Science and High Technology and Environmental
Sciences, Graduate University of Advanced Technology, Kerman, 7631133131, Iran.
*1. r.hassanzdeh@kgut.ac.ir, 2. mehonarmand167@yahoo.com, 3. mh.hosseinjani@gmail.com,*
*4. fnnaseri@yahoo.com*
Correspondence to: Reza Hassanzadeh (*r.hassanzdeh@kgut.ac.ir*).
**Abstract**
This paper proposes a new model in evaluating seismic microzonation of ground shaking by considering
direct characteristics of influencing criteria and dealing with uncertainty of modelling through production
of fuzzy membership functions for each criterion. The relevant criteria were explored by reviewing
previous literature and interviewing 10 specialized experts. Analytic Hierarchy Process (AHP) and Fuzzy
Logic (FL) methods were applied in order to define priority rank of each criteria and to fuzzify sub-criteria
of each criterion by interviewing 10 experts, respectively. Applying Fuzzy Logic method to deal with
uncertainties of sub criteria of each criterion and using direct characteristics of each criterion are the new
approaches in designing a new model. The criteria and sub-criteria were combined in GIS to develop a
model for assessing microzonation of ground shaking in the study area of Bam city, Iran. The model's
output shows high to very high ground shaking levels were happened in central, east, and northeast to
north part of the area. The validation results based on overall accuracy and Kappa statistics showed 80%
to 82% accuracy, 0.74 and 0.75 Kappa indicating a good fit to the model's output. This model assists
planners and decision makers to produce seismic microzonation of ground shaking to be incorporated in
designing new development plans of urban and rural areas, and to facilitate making informed decision
regarding safety measures of existing buildings and infrastructures.
*Keywords: Seismic Microzonation, Site Effects, Ground Shaking, Spatial Modelling, Analytic Hierarchy*
*Process, Fuzzy Logic and GIS.*



## 1. Introduction

This paper explores direct characteristics of influencing criteria and dealing with uncertainty of
modelling through production of fuzzy membership functions for each criterion for the assessment
of ground shaking amplification in a study area. MERM microzonation manual (2003) sets
different factors effecting on the amplitude and duration of ground shaking at a specific site.
These include "the magnitude of the earthquake, focal point and depth of the earthquake,
directivity of the energy release, distance of rapture from the site, geological condition from the
site to the location of the earthquake, and local geology and topographical condition of the site"
(SM Working Group, 2015;Boore, 2003;Hassanzadeh et al., 2013). It has long been known that
local conditions of foundation soils have a significant impact on the effects of an earthquake, as
it was demonstrated in previous earthquakes such as Mexico City, 1985 (Beck and Hall, 1986),
Kobe, 1995 (Wald, 1996), Izmit, 1999 (Tang, 2000) and Umbria-Marche earthquake, 1997 (Moro
et al., 2007). It was witnessed in the Bam earthquake, 2003 that buildings located on
unconsolidated sediments had greater destruction levels (Ramazi and Jigheh, 2006). The aim of
seismic microzonation studies is to prepare ground-shaking map that can communicate efficient
data to planners and policy makers in a geographic area for making informed decision regarding
development policies in urban areas. Therefore, this community require accurate and certain
information for developing mitigation plans and strategies. In the spite of this, there are
uncertainties in estimating seismic microzonation of ground shaking at a site, as this can be
influenced by complex factors such as the estimates of earthquake source, wave propagation, and
site condition. Uncertainty in these criteria results in uncertain ground-motion estimate from
earthquakes (Wang et al., 2017;Wang et al., 2016;Petersen et al., 2016).
Probabilistic Seismic Hazard Analysis (PSHA) (Cornell, 1968) has been used to assess ground-
motion hazards from earthquakes (Atkinson et al., 2015;Petersen et al., 2016). This method
dependent on "the length of the causative faults and depth of the earthquake", which are generally
unknown that cause uncertainty in assessing ground-motion of earthquakes (Wang et al., 2017).
In deterministic seismic hazard analysis (DSHA) (Campbell, 2003;Atkinson and Boore, 2006)
absent of relevant ground-motion attenuation relationship for specific geographic areas can cause
uncertainty in applying DSHA for assessing ground motions of an earthquake (Wang et al.,
2017). Scenario-based seismic hazard analysis (SSHA) (Panza et al., 2012) applies ground-
motion simulations of a scenario earthquake using specified source, path and site parameters. By



conducting many simulations, earthquake variability of different sources, ground-motion
propagation characteristics, and local site effects can be considered. Therefore, uncertainties
using SSHA are quantified explicitly (Wang et al., 2017).
Accurate measurement and communication of uncertainties are critical in ground-motion hazard
assessment for earthquakes. Thus, other approach in microzonation studies is the use of multi-
criteria decision-making methods (MCDM). According to these methods after identifying
potential criteria, experts evaluate and choose among qualitative and quantitative criteria. Since,
experts' judgments can be subjective and imprecise; uncertainty also exists in the analysis.
Uncertainty stems mainly from sources such as the lack of the incomplete data availability,
vagueness, and linguistic expert view. Such uncertainties and vagueness can be dealt with fuzzy
logic principles (Zadeh, 1965) and inference systems  (Klir, 2004;Zadeh, 1975). Based on fuzzy
logic method, the content of each sentence implies logical rules, which constitute the foundation
of fuzzy system modeling and inference procedures. In comprehensive decisions, an expert's
heuristic knowledge or empirical information is used frequently for better conclusions. For these
reasons, Fuzzy Logic is used for evaluating of seismic microzonation of ground shaking
amplification.
There are many MCDM tools in the literature but Analytical Hierarchal Process (AHP) (Saaty,
1980) is one of the most useful techniques, and plays an important role in calculating criteria's
weights and selecting optimized alternatives. Sitharam and Anbazhagan (2008) applied AHP and
GIS for seismic microzonation studies in Bangalore, India. Furthermore, AHP and GIS was
applied to produce seismic microzonation map of Dehli (Mohanty et al., 2007), Haldia, Bengal
Basin (India) (Mohanty and Walling, 2008), Erbaa (Turkey) (Akin et al., 2013) and Al-Madinah
(Moustafa et al., 2016). Fuzzy Logic method was used for evaluation of earthquake damage to
buildings (Sen, 2010), and quick seismic microzonation (Teramo et al., 2005;Nath and
Thingbaijam, 2009;Boostan et al., 2015). Although there were a number of publications
evaluating the seismic microzonation of ground shaking amplification in the literature, but there
is lack of evidence in using the Fuzzy Logic method for producing seismic microzonation of
ground shaking amplification. Moreover, few researchers have considered direct characteristics
of each criteria in local ground shaking analysis. Additionally, in order to remove uncertainties
regarding source of probable earthquake, magnitude and rapture length, therefore these criteria
was not considered for producing seismic microzonation of ground shaking in this study.



The main purpose of this paper is to develop a model for evaluation of seismic microzonation of
ground shaking amplification using AHP, Fuzzy Logic and Weighted Linear Combination
(WLC) methods in GIS. At this stage, model inputs are direct characteristics of local geology,
hydrology, sedimentology, and topographical factors that should be taken into consideration.
First all selected criteria were weighted using AHP method by interviewing 10 experts, then all
criteria are converted into fuzzy sets and fuzzy membership functions (MFs) were produced, then
WLC and fuzzy inference rules are used to develop a model for producing seismic microzonation
of ground shaking amplification for a study area.


**2. Material and methods**
This study investigates the importance of influencing factors on seismic microzonation of ground
shaking. These criteria are identified by reviewing previous literature. Analytic Hierarchy
Process (AHP) and Fuzzy Logic (FL) Methods are applied to deal with selection, weighting and
fuzziness of criteria due to associated uncertainties in the decision-making process of seismic
microzonation of ground shaking amplification by interviewing experts. Combining the criteria
and sub criteria is done based on WLC method. Finally, the developed model is validated using
Overall Accuracy (OA) and Kappa statistics methods. The study has been conducted in four steps
that are elaborated in Figure 1.



Figure 1. The methodological approach of the model



**2.1. Identification, Weighting and Fuzzification of Criteria**
Seismic microzonation of ground shaking can be influenced by several criteria. These criteria
need to be identified by reviewing literature and interviewing experts in data gathering step.
Selected criteria will be weighted and fuzzified using AHP and FL methods as they are explained
in the following:






### 2.2.1. Analytical methods

### 2.2.1. Analytic Hierarchy Process (AHP) method

Several methods have been developed to deal with ranking of criteria and solving a problem,
such as Regime (Hinlopen et al., 1983), ELECTRE family (Figueira et al., 2005), Analytical
Hierarchy Process (AHP) (Saaty, 1980), and Multiple Attribute Utility approach (MAUT)
(Keeney and Raiffa, 1993). AHP is one of the most commonly used multi-criteria decision
making (MCDM) tools, and allows the consideration of both objective and subjective factors in
ranking alternatives in a hierarchical decision model (Saaty, 1980;Saaty, 1990). This method is
applied to convert the experts' view on the importance of each criterion and sub-criterion to a
numerical value  by comparing them to one another, one pair at a time (pair-wise comparison)
(Saaty, 1980).

AHP matrix (A) is developed from the pair-wise comparison of the relative importance of
criterion $A_i$ to criterion $A_j$ ($\alpha_{ij}$, represents a quantified judgment on a pair of criteria $C_i$, $C_j$) (Figure
2), as it was explained above. The values assigned to $\alpha_{ij}$ according to the Saaty's scale (1980) are
usually in the interval of 1 to 9 or their reciprocals. In order to calculate the priority ranking of
each criterion (weight), Saaty (1990) suggested the mathematical computation of eigenvector
(Eq. 1& 2).


Figure 2.AHP matrix (A)


$$\lambda_{max} = \sum_{j=1}^{n} a_{ij} \frac{wj}{wi} \qquad \qquad (Eq.\ 1)$$

Where: $\lambda_{max}$= the largest eigenvalue; $\alpha_{ij}$= judgment; $W_i$ & $W_j$ = numerical weights for judgment
$\alpha_{ij}$.

$$(A - \lambda_{max} I )X = 0 \qquad \qquad (Eq.\ 2)$$

Where: A= AHP matrix; $\lambda_{max}$= the largest eigenvalue; I= Unique matrix; X= eigenvector.






In addition, the assignment of weights (the degree of importance) to each criterion relates to the
process of the experts' logical and analytical thinking, which is tested for each matrix with
Consistency Ratio (CR) statistics. In case, this statistics is less than 0.1 (CR < 0.1) the experts'
answers are logical. Following the testing for consistency, the weights are aggregated to
determine ranking of decision alternatives (the weights) for each criteria. Therefore, in this
research, AHP method is applied to calculate the degree of importance of each criterion
influencing on seismic microzonation level of ground shaking in a region using interview data of
10 specialized experts in seismology, earthquake engineering, geology, tectonics and structural
engineering.



**2.2.2. Fuzzy Logic (FL) method**
Fuzzy logic is a method of "approximating modes of reasoning" (Novák et al., 2012), and it is a
mathematical tool that deals with uncertainty in a different way that can relate independent to
dependent variables. Zadeh (1965) introduced Fuzzy set theory Indicating that the boundary is
not precise and the gradual change is expressed by a membership function, and it changes from
non-membership to membership in a fuzzy set (Eq. 3). The characteristic function can be
assigned a value between 0 to 1. Each membership function is represented by a curve that
indicates the assignment of a membership degree in a fuzzy set to each value of a variable. Curves
of the membership functions can be linear, triangles, trapezoids, bell-shaped, or have more
complicated shapes (Figure 3) depend on the purpose of the subject (Demicco and Klir, 2003).

$A_a = \{x \, \mathcal{E} \, X \mid \mu_A(x) \geq a\}$ *(Eq. 3)*
Where $A_a$ is called the a-cut or a-level set of A, and $\mu_A(x)$ represents membership degree of the
element x.

Figure 3. Fuzzy membership functions (After Mancini, 2012)





Fuzzy systems are mainly based on expert knowledge to formalize reasoning in natural language
mostly using sets of fuzzy inference rules or "*if–then*" rules (Eq. 4).

*If  x is A then y is B*                                    *(Eq. 4)*

As membership functions curve can easily be changed by small increments based on expert
knowledge, therefore, fuzzy logic can characterize and model geologic systems in an efficient
way (Klir, 2004;Demicco and Klir, 2003). Therefore, in this research using Fuzzy set, the
uncertainties in producing microzonation map of ground shaking can be managed by defining
fuzzy membership functions for each criterion. This happens by assigning meaningful values (0
to 1) to each individual (sub criteria) of each criterion through interviewing 10 specialized
experts. For the purpose of defuzzification, largest of maximum method was used that the precise
value of the variable output is one of which the fuzzy subset has the maximum truth-value
(Mancini et al., 2012).

## 200    2.3.    Data gathering

In order to identify influencing criteria in seismic microzonation of ground shaking the required
data were collected through a literature review, and semi-structured interviews with 10 experts
who were involved in the geology, seismology, tectonic and structural engineering, and
geomorphology fields. They were asked about the criteria that can influence seismic microzo
nation level of ground shaking, and then these data were analyses using AHP and FL methods as
explained in the following:

### 208    *2.3.1.   Determining the relevant criteria by reviewing literature*

The potential criteria influencing seismic microzonation of ground shaking were determined
through reviewing previous research. By reviewing documents on earthquake engineering,
seismology, geology, tectonic and structural engineering, geomorphology and seismic
microzonation reports and guidelines (Fäh et al., 1997;Ding et al., 2004;Molina et al.,
2010;Mundepi et al., 2010;Marulanda et al., 2012;Hassanzadeh et al., 2013;Federal Emergency
Management Agency (FEMA), 2014;Fraume et al., 2014;Grelle et al., 2016;Grelle et al., 2014;SM





Working Group, 2015;Rehman et al., 2016;Nwe and Tun, 2016;Global Earthquake Model (GEM), 2017;CAPRA, 2017;Michel et al., 2017;Trifunac, 2016;Hassanzadeh and Nedovic-Budic, 2016), in total 14 criteria were recognized that can influence seismic microzonation levels in a study area (Table 1).

Table 1.Relevant criteria that influence on seismic microzonation

### 2.3.2. Experts' Knowledge data

a) *Interviewing disaster managers (semi-structured interviews) to determine the important criteria*

The most important criteria were determined by conducting a semi-structured interview with 10 experts using the snowball sampling or chain-referral sampling method (Biernacki and Waldorf, 1981). In this study, all 10 interviewees were highly experienced and had been involved in seismic microzonation studies. The average age of the sampled individuals was 43 years, and all of them had a postgraduate degree.

A list of criteria that were identified by reviewing previous studies were given to the experts and they were requested to add other criteria if they thought they were applicable. They were asked to rank each criterion using a five-point Likert Scale (Likert, 1932), so respondents could choose the option that best reflected their opinion on each criterion. When surveying many people on the same criterion, the five codes could be summed up, averaged or calculate the mode, indicating overall positive or negative orientation towards that criterion. This was the basis from which this method was used to identify the degree of importance for each criterion in seismic microzonation of ground shaking in a region. Therefore, in order to elicit the most relevant criteria, the significance of specific factors were measured on a five-point Likert Scale where, 1 represents 'not important at all', 3 'of little importance', 5 'of average Importance', 7 'very important', and 9 'extremely important' (Likert, 1932;Jamieson, 2004). The collected data from experts were analysed and criteria with mean ratings above '5' ('of average important') were selected (Table 2). These are considered for further analysis using the Analytic Hierarchy Process (AHP) method.




Table 2.The average importance criteria based on 5-point Likert Scale

***b) Interviewing disaster managers (structured interviews) in order to collect data for***
***computing the relative importance (weights) of the criteria***
A questionnaire based on AHP matrix (A) was developed for a pair-wise comparison of the
relative importance of the criteria for calculating the weights (priority ranking) of each criterion.
As AHP is a subjective method therefore a large sample size is not needed (Cheng and Li,
2002;Lam and Zhao, 1998). Therefore, data were collected by interviewing 10 experts (the same
experts who were interviewed in the first round) based on the structured questionnaire (closed-
ended questions). They were asked to compare the relative importance of each criterion against
all others, based on Saaty's scale by verbal preferences (Saaty, 1980). A pair-wise comparison
that was carried out with an expert is shown in Table 3. These data are used by the AHP method
to compute the weight of each criterion as explain previously.

Table 3.The results of pair-wise comparisons of the selected criteria with each other based on
the AHP matrix


***c) Determining fuzzy set and fuzzification of thresholds of sub-criteria for each criterion***
In the next step, since each criterion and its sub-criteria has different effect on the seismic
microzonation of ground shaking level in a region, fuzzy membership functions (MFs) for sub
criteria of each criterion are defined in that numerical analyses of their effect would be computed.
As, designed parameters of each membership function depends on experts knowledge, then
number of memberships, the shape, the positioning, and the overlay area of memberships of each
MFs for each criterion would be different. To conduct this analysis, 10 experts were interviewed
regarding membership degree of sub criteria of each criterion, and mode of each sub criteria was
calculated and MFs for each criterion was depicted as descried in the following:
- Thickness of soil and sediments: an effective factor in site effect assessment is the thickness of
sediments. Rezaei et al. (2009) (2009) state that the soil thickness shows a direct relationship to




damage rate observations in the Bam earthquake. This layer was produced by 245 geophysical,
geotechnical, and sedimentological sample sites across the city. The alluvial thickness varies in
different parts of the city. In the northern part of the city, the sediment thickness ranges from 0 m,
where bedrock is exposed beneath Arg-e-Bam, to 90 m across most of the northern half of the
study area. Toward the south and center of the study area, sediment thickness increases over a
short distance, to more than 270 m. This defines a subsurface of high sediment thickness that
extends across the entire study area from west to east and underlies south-central Bam. Therefore,
based on a direct relationship between the damage rate and alluvial thickness (Rezaei et al.,
2009;Marie Nolte, 2010). MF for this criterion is depicted in figure 4a.

- Consolidation and strength of soil and sediments: It has been frequently observed that earthquake
damage is greater in settlements located on unconsolidated and soft soils than in those sited on
stiff soils or hard rock. For example, in Bam earthquake strong amplification occurred due to the
extremely soft clay layers that caused high-rise buildings to collapse (Jafari et al., 2005). Another
example was the Loma Prieta earthquake that happened in 1989, where much of the damage
occurred in the central San Francisco Bay area at sites underlain by thick deposits of soft clay soils
(Stewart, 1997). The soil classification has been done based on different thresholds for the average
shear wave velocity (Vs) to a depth of 30m by the National Earthquake Hazard Reduction Program
(NEHRP) to characterize sites for purposes of estimation amplification of seismic motions. This
standard has applied in Unified Building Code (Dobry et al., 2000) and Eurocode8 (Sabetta and
Bommer, 2002;Kanlı et al., 2006). Based on this classification in areas on unconsolidated
sediments, shear wave velocity reduces, and expected amplification during earthquakes cab be
increased. Therefore, according to this MFs for each class have been calculated as shown in figure
4b.

- Type of soil and particle size distribution of sediments: It has long been recognized that the
destructiveness of ground shaking during earthquakes can be significantly worsened by the type
of local soil and subsurface sediment conditions. In past events, the observed variability in seismic
intensity and structural damage severity has often been attributed to the variability of soil and
subsurface sediment stratigraphy in a given area. Among the geotechnical properties of soil and
sediments, grain size is one of the most important criteria (Assimaki et al., 2006;Phoon et al.,





2006). In the study area, Rezaei et al. (2009) identified eight sediment types: clay, silt, sand,
granules, pebbles, cobbles, and boulders. They stated that the grain size in the shallow subsurface
(<10 m) decreases across the city from south to north and increases with depth. Their investigation
showed that fine-grained soils and sediments (clay, clayey sand, cohesive sandy mud, and cohesive
muddy sand) dominated the northern part of the city at shallow depths. In the central part of the
city, fine-grained sediments changed laterally to coarse-grained sediments (poorly sorted sand,
well-rounded gravel, poorly sorted gravel, and muddy or sandy gravel) which dominated in the
south part of the city. As a rule, it can be assumed that, the smaller the grain size of sediments, the
less the shear waves velocity and therefore the greater the effect of the seismic wave on the
destruction level of buildings (Rezaei et al., 2009;Assimaki et al., 2006;Phoon et al., 2006).
Therefore, the MFs for each specific grain size are calculated in Figure 4c.

- Depth of groundwater: Research on the effects of groundwater shows it can magnify an
earthquake's damage. The most well known effect is liquefaction. The geologic and hydrologic
factors that affect liquefaction susceptibility are the age and the type of sedimentary deposits, the
looseness of cohesions less sediments and the depth to the groundwater table (Tinsley et al., 1985).
The liquefaction is mostly limited to water-saturated, cohesions less sediments, and granular
sediments at depths less than 15m (Iguchi and Tainosho, 1998;Sitharam, 2010). Noack and Fah
(2001) categorized it by the depth of the water table, which is split into three classes where the
weight of the class increases while the groundwater table decreases (Fah et al., 1997). Therefore,
due to the geological conditions in Bam, liquefaction is considered of minor importance because
Talebian et al. (2004) and Rezaei et al. (2009) found water saturated sands in very few places,
however, measured microtremore data demonstrated more applification in areas with high
groundwater levels. Accordingly, MFs for each class of groundwater depth are computed as shown
in figure 4d.
- Type of surficial rock: Type of surficial rocks can effect on seismic microzonation level of ground
shaking in each region. Three main types of rock based on their formation process include igneous,
metamorphic, and sedimentary rocks. Each type has its own sub-categories and what matter in this
research is how hard or soft and how dense the specific type of rock is in compare with the other
types. Geological Strength Index (Geological Survey of Iran (GSI)) of "rock masses depends on





rock's material, the amount of joints and their relations, alteration, and presence of water" (Hoek
and Brown, 1997). There are many rock types in the nature that GSI can be calculated for any of
them based on their condition, and then can be fuzzified addressing their effect on seismic
microzonation level of ground shaking. There are five classes of GSI including very good, good,
fair, poor and very poor based on their surface quality and interlocking of rock pieces from
massive, blocky, very blocky, disintegrated, and laminated/ sheered (Marinos et al., 2007). The
GSI values categorized in five classes including very low, low, medium, high and very high levels.
These classes shows the geological strength of rocks that the high and very high GSI demonstrate
high to very high strength of rocks. Therefore, previous studies demonstrates that in massive rocks,
high GSI values, seismic waves passes quickly and therefore have small influence in seismic
microzonation level of ground shaking, and vice versa if GSI value gets to the lower values. Thus,
in fuzzyfication process of surficial rocks, the rock with very high GIS assign 0 and the rocks with
very low GSI assign 1 (Figure 4e). Furthermore, the criterion of type of bedrock acts the same as
surficial rock type criterion as explained above. Type of bedrock rarely changed over a small extent
with homogenous lithology. However, it was concern of experts in determining seismic
microzonation of ground shaking.
- Slop surface: Bisch et al. (2012) reported that the effects of slope angle on topographic
amplification factor. They classified the slop angle to three categories: 0-15 with no effect, 15-30
degree with 1.2 and more than 30 degree with 1.4 amplification coefficients. Bouckovalas and
Papadimitriou (2005) investigated that the influence of slope in amplifying the peak horizontal
seismic ground acceleration in front and behind the crest. Grelle et al. (2016) presented formulae
for topographic amplification on slope surface. These studies indicated that with the increase in
slope angle the amplification factor would be increased. This can be a basis for depicting MFs of
this criterion (Figure 4f).
- Topography irregularities: Seismic amplification has been witnessed in several earthquakes due
to topographical changes (Geli et al., 1988;Paolucci, 2002). Bisch et al. (2012) classified the site
in two classes of "isolated cliff and ridge with crest width significantly less than base width" (CEN
European Committee for Standardisation, 1994, p 93). However, this seems simplistic, as it does
not consider the elevation differences. Furthermore, Grelle et al. (2016) presented an equation that
considered the local slope height, relief height, regional share wave velocity and relief ratio. In



addition, several calibration constants should be calculated using 2d numerical analysis for each
study area to compute topographic effects on seismic microzonation of ground shaking. Lee et al.
(2009) found out that the amplification on top of elevated surfaces with small extent was much
higher than valleys and flat areas. Therefore, the elevation differences (dH m) between the bases
of a hill with the top of the hill and also the area (A m$^2$) of the top part of the hill are the main
driver in computing the amount of amplification of seismic waves and can effect on seismic
microzonation level of ground shaking. Therefore, the higher the elevation differences and the
smaller the area of the elevated surface, the ground in this part will be more amplified. Here, using
fuzzy logic and experts' knowledge the effect of topography in terms of elevation differences in
determining seismic microzonation of ground shaking in the study area is defined (Figure 4g).
Figure 4. Membership functions (MFs) based on fuzzy logic system: Thickness of soil and
sediments (a), Consolidation and strength of soil and sediments (b), Type of soil and particle
size distribution of sediments (c), Depth of groundwater (d), Type of surficial rock and
bedrock (e), Slope surface (degree) (f), Topography irregularities (g).

### 2.3.3. Preparing thematic data

The required data were collected from relevant organizations and documents and they were
converted to GIS files. These thematic data included: thickness of soil and sediments (Figure 5a),
consolidation and strength of soil and sediments (Figure 5b), type of soil and particle size
distribution of soil and sediments (Figure 5c and d), depth of groundwater (Figure 5e), type of
surficial rock (Figure 5f), topography of surface (Figure 5g), and slop surface (Figure 5h) layers.
Figure 5. Thematic Layers of Bam city: Thickness of soil and sediments (m) (a), Consolidation
and strength of soil and sediments, (b), Sediment type at depth of 1 meter (c) and at depth of 9
meters (d), Groundwater level (e), Type of surficial rock(f), Topography (g) and Slop (h) layers.

### 2.3.4. Preparing control data

National Cartographic Center (2003) and Hisada et al.(2005) were collected data on the destruction
level of buildings after math of the bam earthquake (Figure 6a and b). Lashkari Pour et al. (2006)





and Askari et al. (2004) were collected data on the dominant frequency of soil (Figure 6c and d)
using microtremor measurements in Bam city. These datasets were used to validate the model.

Figure 6. Control data: Actual building destruction level (Hisada et al., 2005) (a), percentage of

damage to buildings caused by the Bam earthquake in 2003 (National Cartographic Center

(NCC), 2003) (b), Dominant frequency by(LashkariPour et al., 2006) (c) and by (Askari et al.,

2004) (d) using Microtremor field measurement.


**2.3. Spatial combination methods and overlay rules**

The spatial Multi Criteria Decision Making (MCDM) approach is a decision-aid and a
mathematical tool that combines and transforms spatially referenced data into a raster layer with
a priority score. (Roy, 1996;Malczewski, 2006). Several combination methods have been
developed, such as Boolean operations (Malczewski, 1999), weighted linear combination (WLC:
combining the normalized criteria based on overlay analysis) (Voogd, 1983;Drobne and Lisec,
2009;O'Sullivan and Unwin, 2010) (Eq. 5), ordered weighted averaging (OWA) (Yager,
1988;Rinner and Malczewski, 2002), and Analytical Hierarchy Process (AHP) based on the
additive weighting methods (Zhu and Dale, 2001). In this research, the AHP method (Saaty,
1980) was used to derive the weights associated with criteria and Fuzzy Logic method was
applied to compute sub-criteria's membership functions (MFs) in order to produce the seismic
microzonation of ground shaking. Then, the degree of membership of each sub-criteria
(calculated by Fuzzy Logic method) is assigned to the corresponding sub-criteria. Next, this is
multiplied by the weight of corresponding criteria (calculated by AHP method). Finally, they are
summed up in a linear manner using WLC method (Eq. 5) to develop the model (Larzesh model)
for production of the seismic microzonation of ground shaking in the study area.

$$Ai = \sum W_j * X_{ij} \qquad\qquad\qquad (Eq.\ 5)$$

*Where: $w_j$ = the calculated weight of criteria j, and Xij = the degree of memebrship of the ith*
*sub-criteria with respect to the jth criteria, and Ai = the seismic microzonation of ground*
*shaking index in ith location.*

**2.4.    Validation and comparison methods**





In order to validate the model, as categorical variables are the main driver of model development in
this research, therefore relevant measures such as Overall Accuracy and Kappa statistic will be
applied to measure the performance of the model.

### a) *Overall accuracy (OA)*

Accuracy assessments determine the quality of the results derived from data analysis or a model,
in comparison with a reference or ground truth data (where ground truth data are assumed to be
100% correct) (Congalton and Green, 2009). The accuracy assessment can be obtained by
creating a contingency table of counts of observations, with calculated, estimated or predicted
data values as rows and with reference data values as columns. The values in the shaded cells
along the diagonal represent counts for correctly classified observations, where the reference data
matches the predicted value. This contingency table is often referred to as a confusion matrix,
misclassification matrix, or error matrix (Czaplewski, 1992;Congalton and Green, 2009) (Eq. 6).

$$OA = \frac{\sum_{k=1}^{q} n_{kk}}{n} \times 100 \qquad\qquad (Eq.\ 6)$$


*Where: OA = Overal Accuracy, $n_{kk}$ = Values in diagonal cell of the matrix (correctly classified*
*observations), and n = number of observations.*

### b) *Kappa analysis*

The kappa statistic (κ) (Sim and Wright, 2005;Congalton and Green, 2008) calculates degree of
agreement between classes of two independent observe measuring the same property. The degree
of Kappa would be 0 for a random classifies and 1 for classification. Degree of agreement of
Kappa interprets as follows: less than 0.4: poor agreement, 0.4 and 0.8:  moderate agreement,
and greater than 0.80: strong agreement (Congalton and Green, 2008) (Eq. 7).

$$k = \frac{P_o - P_e}{1 - P_e} \qquad\qquad (Eq.\ 7)$$
*Where: Po = the relative observed agreement among raters, Pe = the hypothetical probability of chance*
*agreement.*



**Results and discussion**

In order to produce the seismic microzonation of ground shaking the most important criteria were
identified and then were weighted using AHP pair-wise comparison method. The higher weight
belong to thickness of soil and sediments (0.271), consolidation and strength of soil and
sediments (0.207), type of soil and particle size distribution of sediments (0.177), depth of
groundwater (0.171), topography of surface (0.054), type of surficial rock (0.041), slop surface
(0.040), and type of bedrock (0.040) were considered. Then, based on Fuzzy Logic method sub-
criteria of each criterion was fuzzified and membership functions for them was defined. Next,
these criteria were combined based on the Weighted Linear Combination (WLC) (Drobne and
Lisec, 2009) in GIS to develop the model for producing the seismic microzonation of ground
shaking map of the study area, as it is proposed in the following (Eq. 8):

$$A_j = \sum(wS_s \cdot FS_{ss}) + (wT_A \cdot FS_{TA}) + (wS_A \cdot FS_{SA}) + (wD_{Gw} \cdot FS_{DGW}) + (wT_R \cdot FS_{TR})$$

$$+ (wT_{Br} \cdot FS_{TBR}) + (wT_S \cdot FS_{TS}) + (wS_L \cdot FS_{SL}) \qquad (Eq.\ 8)$$

*Where: $A_j$ = seismic microzonation of ground shaking, weights of each criterion: $wS_s$ = consolidation
and strength of soil and sediments , $wT_A$ = thickness of soil and sediments, $wS_A$ = Type of soil and particle
size distribution of sediments , $wD_{Gw} = depth\ of\ groundwater$ , $wT_R$ = type of surficial rock , $wT_{Br} =$
type of bedrock, $wT_S$ = topography of surface, $wS_L$ = slop surface, and fuzzified sub-criteria of each
criterion: $FS_{ss} = consolidation\ and\ strength\ of\ soil\ and\ sediments$, $FS_{TA}$ = thickness of soil and sediments
, $FS_{SA} = Tyep\ of\ soil\ and\ particle\ size\ distribution\ of\ soil\ and\ sediments$, $FS_{DGW} =$
$depth\ of\ groundwater$, $FS_{TR}$ = type of surficial rock , $FS_{TBR}$ = type of bedrock, $FS_{TS}$ = topography of
surface , $FS_{SL}$ = slop surface.*

Figure 7 displays the resulting microzonation map of ground shaking in Bam city. The areas with
high to very high susceptibility of amplification are located in the north, east and northeast part
of Bam city. This is due to the widespread unconsolidated sediments, low groundwater level in
combination with high sediment thickness.

In order to validate the results OA and Kappa methods were applied comparing the output of
model with the measured predominant frequency (Askari et al., 2004;LashkariPour et al., 2006)
in the study area. The results demonstrated 80% and 82%  (Table 4a and b) for OA and 0.74 and



0.75 for Kappa (Table 5) indicating a good fit of the model's output with the measured data.
Moreover, overlaying the building destructions caused by the Bam earthquake in 2003 (Hisada
et al., 2005;National Cartographic Center (NCC), 2003) shows high destruction levels happened
in locations with high ground shaking which were located in central, north and northeast part of
the city.

Figure 7. Seismic microzonation of ground shaking map of Bam city

Table 4. Coparesion between the model's output with the measured predominant frequanecy in Bam
city by Askari et al. (2004) (a)c and LashkariPour et al. (2006) (b).

Table 5. Kappa coefficient and OA

In this study, we have focused on the site effect and local geology properties of a site that have a
massive influence on seismic microzonation of ground shaking in the study area. To deal with
related uncertainties in preparing seismic microzonation, the most important criteria were selected,
weighted and the fuzzified. Criteria with high uncertainty degree such as distance of active fault
to the site, depth and magnitude of the probable earthquake were not considered because there was
no possibility to exactly find out where and how an earthquake will be triggered. Therefore, only
the criteria with known location (x and y) and known characteristics were taken into consideration.
Furthermore, to deal with uncertainties Fuzzy Logic is a suitable approach as we can define
membership function of the effect of each criterion in the amplification of ground shaking by
interviewing 10 experts and obtaining expert's knowledge. This can result in realistic output
regarding the behavior of each criterion in ground shaking calculation.
The newly developed model uses AHP and Fuzzy Logic (Zadeh, 1965) to deal with complexities
and uncertainties in data analyses in weighting the criteria and fuzzifying the sub-criteria of each
criterion. Although, in studies for evaluating seismic microzonation in Bangalore (India) (Sitharam
and Anbazhagan, 2008), Dehli (Mohanty et al., 2007), Haldia (India) (Mohanty et al., 2007), Erbaa
(Turkey) (Akin et al., 2013) and Al-Madinah (Moustafa et al., 2016) only AHP method was applied
to weight the criteria, and none of these studies considered weighting of sub criteria for each
criterion even using other methods.



Few researchers have considered direct properties of influencing factors in assessing ground
shaking amplification. Even, in evaluating seismic response developed models such as
SiSeRHMap v1.0 (Grelle et al., 2016) and GIS Cubic Model (Grelle et al., 2014), the researchers
have applied only lithodynamic, stratigraphic and topographic effects as influencing factors. The
current research considers direct properties of each criteria and tries to manage uncertainties in
criteria and sub-criteria of each criterion via weighting and fuzzification process using experts'
knowledge and the use of direct properties of criteria. These processes can be extended in more
details, which are subject to more investigation in the future.

**Conclusions**
Larzesh model introduces a new method based on AHP and Fuzzy Logic rules that enables experts
to produce seismic microzonation of ground shaking using direct properties of lithological,
sediment-logical, geological, hydrological and topographical effects in a study area using experts'
knowledge in weighting and fuzzifing criteria and sub criteria that can be readily perceived and
consulted.
The application of the model was carried out in the urban area of the Bam city in Iran. The results
demonstrated high to very high ground shaking amplifications were located in central, east, and
northeast to north part of the city that was confirmed comparing with measured microtremor data
on predominate frequency in the study area. However, as the proposed model is a spatial
computational tool, the validation of output in producing seismic microzonation of ground shaking
strictly dependent on the quality and preparation of input data.
In conclusion, the model enable disaster managers, planners, and policy makers in producing
seismic microzonation of ground shaking and making informed decision in urban planning and
designing appropriate plans for urban development, especially in areas with high seismic activities.

**Acknowledgements**
The authors would like to express their appreciation to Institute of Science and High Technology
and Environmental Sciences, Graduate University of Advanced Technology, Kerman, Iran for
financial support of this study under reference number of 7/C/95/2053.



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





**Tables**


Table 1.Relevant criteria that influence on seismic microzonation

| 1 | Thickness of soil and sediments | 9 | Thickness of bedrock |
|---|---|---|---|
| 2 | Consolidation and strength of soil and sediments | 10 | Morphology of bedrock |
| 3 | Type of soil and particle size distribution of sediments | 11 | Topography of bedrock |
| 4 | Depth of groundwater | 12 | Age of alluvial and sediments |
| 5 | Topography of surface | 13 | Age of bedrock |
| 6 | Type of surficial rock | 14 | Age of surfacial rock |
| 7 | Slop surface | | |
| 8 | Type of bedrock | | |


Table 2.The average importance criteria based on 5-point Likert Scale

| | Criteria for | Average degree of |
|---|---|---|
| 1 | Thickness of soil and sediments | 8.5 |
| 2 | Consolidation and strength of soil and sediments | 8 |
| 3 | Type of soil and particle size distribution of sediments | 7.5 |
| 4 | Depth of groundwater | 7.25 |
| 5 | Type of surficial rock | 7 |
| 6 | Topography of surface | 5.25 |
| 7 | Slop surface | 5 |
| 8 | Type of bedrock | 5 |
| 9 | Thickness of bedrock | 4.5 |
| 10 | Morphology of bedrock | 4.5 |
| 11 | Topography of bedrock | 4.5 |
| 12 | Age of alluvial and sediments | 3.75 |
| 13 | Age of bedrock | 3.25 |
| 14 | Age of surfacial rock | 2.75 |





Table 3.The results of pair-wise comparisons of the selected criteria with each other based on
the AHP matrix

| Criteria | 1 | 2 | 3 | 4 | 5 | 6 | 7 | 8 | Weights |
|---|---|---|---|---|---|---|---|---|---|
| 1-Thickness of soil and sediments | 1 | 1 | 2 | 2 | 5 | 5 | 7 | 4 | 0.271 |
| 2-Consolidation and strength of soil and sediments | | 1 | 1 | 1 | 5 | 4 | 5 | 5 | 0.207 |
| 3-Type of soil, and particle size distribution of sediments | | | 1 | 1 | 5 | 5 | 5 | 7 | 0.177 |
| 4-Depth of groundwater | | | | 1 | 5 | 7 | 3 | 5 | 0.171 |
| 5-Type of surficial rock | | | | | 1 | 2 | 1/2 | 1/2 | 0.041 |
| 6-Topography of surface | | | | | | 1 | 1/2 | 3 | 0.054 |
| 7-Slop surface | | | | | | | 1 | 4 | 0.040 |
| 8-Type of bedrock | | | | | | | | 1 | 0.040 |
| Lambda = 8.60    CI = 0.05 | | | | | | | | | |

Table 4. Coparesion between the model's output with the measured predominant frequanecy in Bam
city by Askari et al. (2004) (a) and LashkariPour et al. (2006) (b).
**a)**

| Predicted | Predominant Frequency ( Measured) | | | | | |
|---|---|---|---|---|---|---|
| | **1** | **2** | **3** | **4** | **5** | **Total** |
| 1 | | | 1 | | | 1 |
| 2 | | 2 | | | | 2 |
| 3 | 1 | | 1 | 1 | 1 | 4 |
| 4 | | | | 7 | | 7 |
| 5 | | | | | 9 | 9 |
| **Total** | **1** | **2** | **2** | **8** | **10** | **23** |
| Av_Ac = 82 % | | | | | | |

**b)**

| Predicted | Predominant Frequency ( Measured) | | | | | |
|---|---|---|---|---|---|---|
| | **1** | **2** | **3** | **4** | **5** | **Total** |
| 1 | 1 | | | | | 1 |
| 2 | | 1 | | | | 1 |
| 3 | | | 3 | | | 3 |

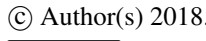



| 4 | | | | 1 | | 1 |
| 5 | | 1 | | | 1 | 2 | 4 |
| **Total** | | **2** | **1** | **3** | **2** | **2** | **10** |
| **Av_Ac = 80 %** | | | | | | | |


Table 5. Kappa coefficient and OA

| Comparison of the model's output and measured data | Predominant frequency (Askari et al., 2004) | Predominant frequency (LashkariPour et al., 2006) |
|---|---|---|
| Kappa coefficient | 0.74 (0.000) | 0.75 (0.000) |
| OA | 82% | 80% |


















**Figures**

1. Identification, Weighting and Fuzzification of Criteria

Reviewing previous literature to detect relevant criteria for seismic microzonation of ground shaking amplification

Interviewing experts to identify the most important criteria

Interviewing experts to calculate weights of selected criteria using AHP method

Interviewing experts to fuzzify the sub-criteria of each criterion using Fuzzy Membership Functions

2. Combining criteria and sub criteria using Weighted Linear Combination (WLC) method to produce seismic microzonation map for the study area.

3. Proposing a spatial model (Larzesh Model) for computing amplification of ground shaking in the case study area

4. Validating the model using predominant frequency of measured Microtremor data and actual building destruction levels caused by the Bam earthquake in 2003

Figure 1. The methodological approach of the model

$$
A = \begin{bmatrix} a_{11} & a_{12} & \cdots & a_{1n} \\ a_{21} & a_{22} & \cdots & a_{2n} \\ \vdots & \vdots & \ddots & \vdots \\ a_{n1} & a_{n2} & \cdots & a_{nn} \end{bmatrix}
$$


Where: $a_{ij} = 1$, if $i = j$, and $a_{ij} = \frac{1}{a_{ij}}$, if $i = \overline{1,n}$ and $j = \overline{1,n}$.

Figure 2. AHP matrix (A)



Figure 3. Fuzzy membership functions (After Mancini, 2012)


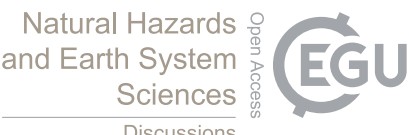

a)
c)
e)





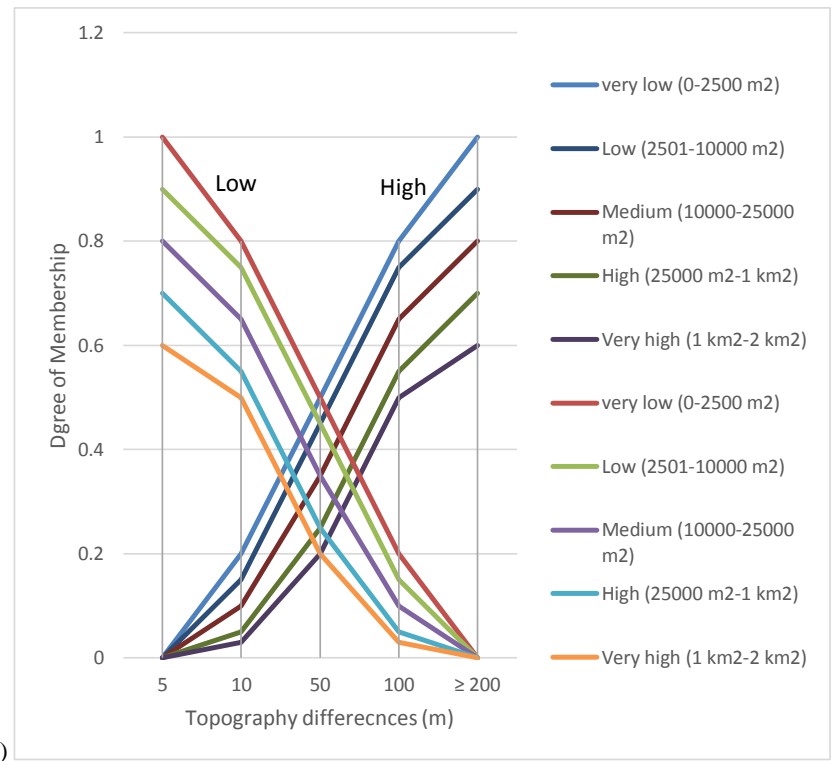

833     g)

834   Figure 4. Membership functions (MFs) based on fuzzy logic system: Thickness of soil

835  and sediments (a), Consolidation and strength of soil and sediments (b), Type of soil and

836    particle size distribution of sediments (c), Depth of groundwater (d), Type of surficial

837     rock and bedrock (e), Slope surface (degree) (f), Topography irregularities (g).


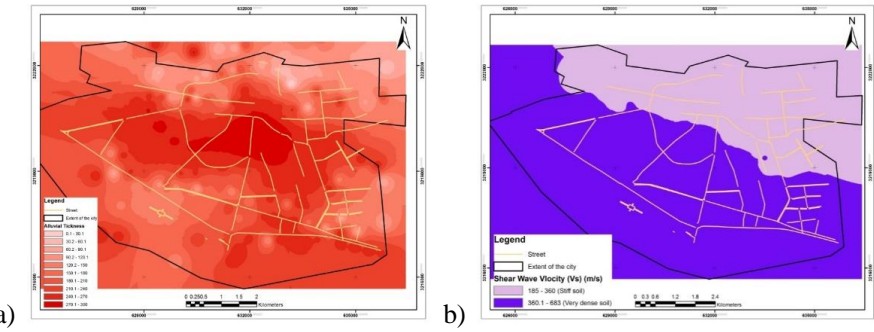

839   a)                b)



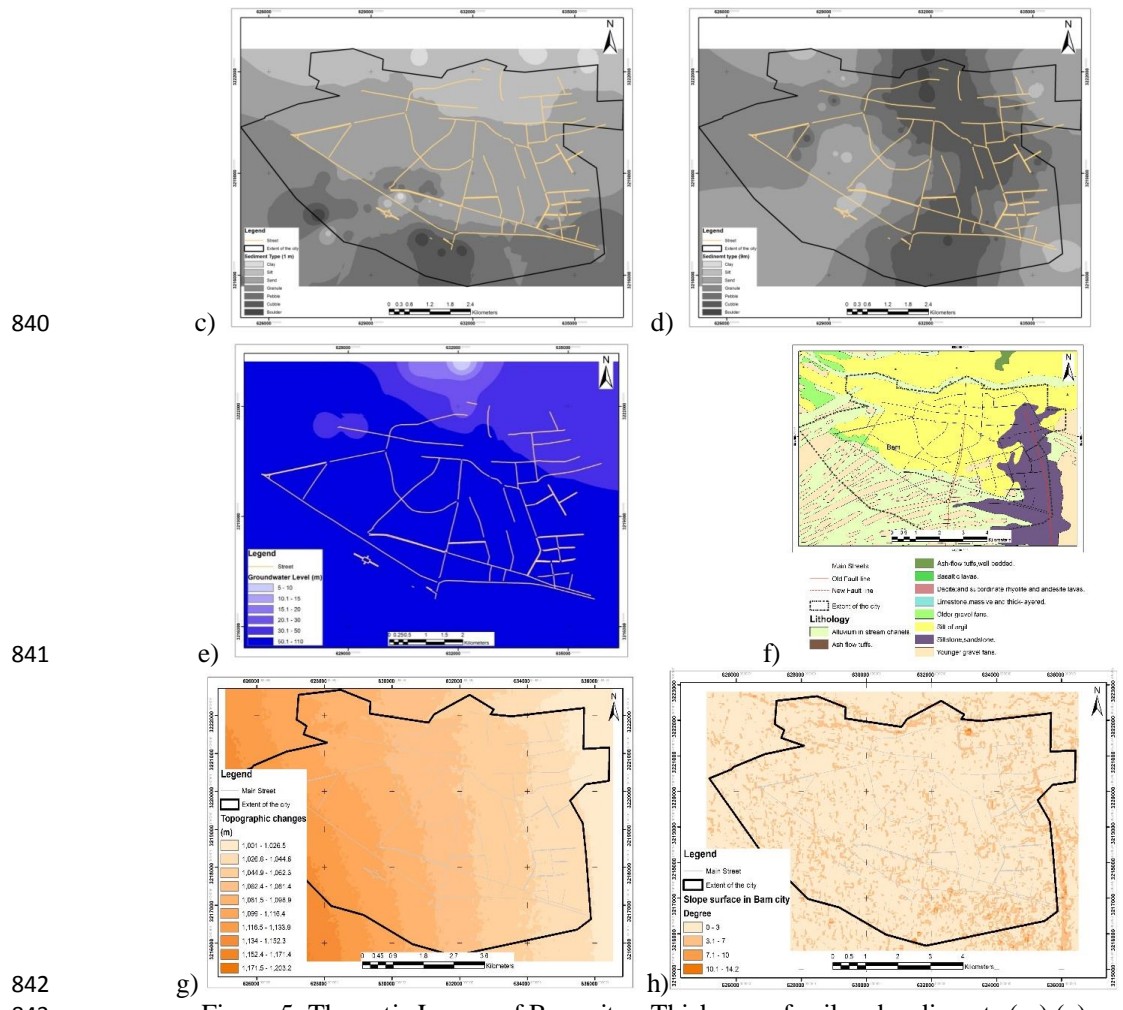

c)                                    d)
e)                                    f)
g)                                    h)

Figure 5. Thematic Layers of Bam city:  Thickness of soil and sediments (m) (a),
Consolidation and strength of soil and sediments, (b), Sediment type at depth of 1 meter (c)
and at depth of 9 meters (d), Groundwater level (e), Type of surficial rock(f), Topography (g)
and Slop (h) layers.






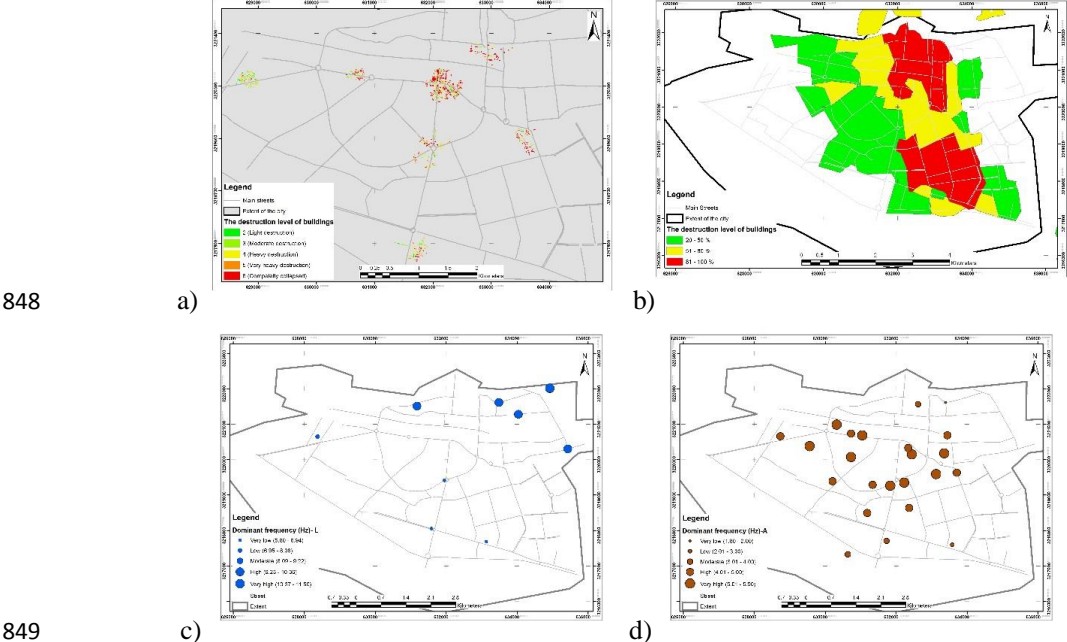

a)                                  b)

c)                                  d)


Figure 6. Control data: Actual building destruction level (Hisada et al., 2005) (a), percentage of
damage to buildings caused by the Bam earthquake in 2003 (National Cartographic Center
(NCC), 2003) (b), Dominant frequency by (LashkariPour et al., 2006) (c) and by (Askari et al.,
2004) (d) using Microtremor field measurement.


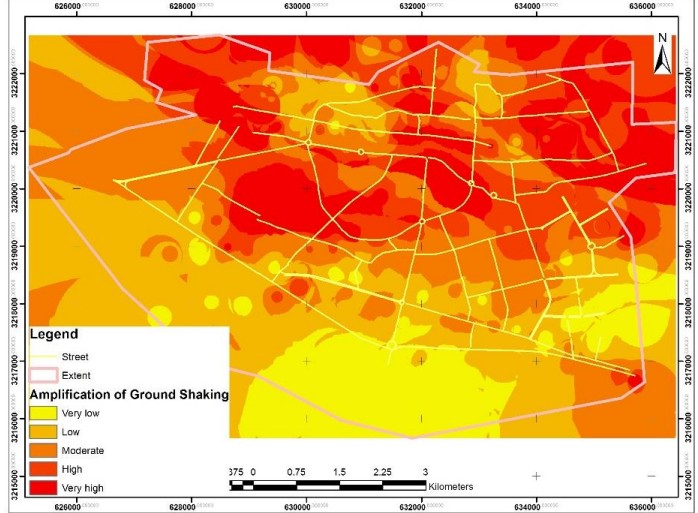

Figure 7. Seismic microzonation of ground shaking map of Bam city