# Peer review of "Natural Hazards"

_Natural Hazards and Earth System Sciences, 2017_

## Referee Comment (RC1) · Anonymous Referee #1 · 5 Oct 2018

The paper presents the application a new methodology for seismic microzonation based on analytical hierarchy process and fuzzy logic methods. The procedure was applied to Bam city and the microzonation map was validated by comparing the output of study with experimental predominant frequencies and with the observed damage pattern. The results are encouraging, and the paper is well written however I have some relevant comments to be addressed by the authors. General comments. 1) The output is a qualitative ranking of the susceptibility of the area to ground motion amplification phenomena. The Authors should stress that is a kind of level-1 or grade-1

microzonation study according to international standards and guidelines. More detailed studies are needed in high-risk areas and should be based on numerical modelling of site effects (i.e, physically based procedures which cannot be replaced by influencing criteria methods). 2) my biggest doubts concern the criteria selected for the procedure. "consolidation and strength": consolidation should be replaced by stiffness (in terms of shear wave velocity or shear modulus) while the strength (i.e. soil resistance) is not pertinent; the resistance should influence the slope stability or the resistance to liquefaction not the amplification phenomena. The "particle size distribution" does not affect the amplification phenomena: the stiffness of soil is the controlling factor not the particle size: a coarse-size if loose soil can be softer than a consistent over consolidated fine grained clay ! The "depth of groundwater" is not pertinent for amplification effects, it only controls the possible occurrence of soil liquefaction in loose sandy soils. In table 1 is not clear the difference between morphology of bedrock and topography of bedrock (even if, as I understand, they are considered as minor criteria). I strongly suggest to remove these factors or better define them. Specific comments 1) It is not clear how the output of model is compared with experimental resonance frequencies. In terms of amplitude of HVSR peak with levels of shaking map ? Please describe better this validation phase. 2) Pag. 3: at the state of art, I don't believe that SSHA methods can be used for microzonation: source and path parameters are still quite significant to define. PSHA assessments at regional scale (at outcropping rock) and site response modelling at local (urban scale) are the most adopted procedure almost worldwide. Typing 3) Pag. 11: "microtremor" instead of "microtremore"

---

## Referee Comment (RC2) · Anonymous Referee #2 · 9 Oct 2018

Dear Editor, I just reviewed the paper entitled "New Approaches to Seismic Microzonation Modelling of Ground Shaking Using Direct Characteristics of Influencing Criteria: Case Study of Bam City, Iran". This paper deals with a new approach for seismic microzonation in urban areas. I think the paper could be accepted for publication only after a major revision. There are several points that need to be fixed. First of all, the paper need a strong revision of the English by an English mother tongue. Second critical point, authors shoud avoid to repeat the same concepts too many times just by using sligthly different sentences. Third, very important point. Authors talk about seismic microzonation of ground shaking amplification. This concept is not clear to me. Do they mean "seismic site response"? This is a crucial point that needs to be clarified. In addition, in the discussion section, authors suddenly introduce the concept of "susceptibilty amplification" (line 469). Susceptibility is different from seismic site response! Authors need to clear state these concepts in the entire paper. Fourth critical point, the method section is 12 pages long whereas the discussion and conclusion section is just 3 pages! This discrepancy is incredible! Methods section include too many repeats of the same concepts (e.g., authors said several time that they interviewed 10 experts!). In additiond, discussion and conclusion section needs to be more detailed. Fifth point, most of the figures are very hard to read. Quality of figures should be increased. Several other comments are listed in the attached file. Best.

Please also note the supplement to this comment:
https://www.nat-hazards-earth-syst-sci-discuss.net/nhess-2017-421/nhess-2017-421-RC2-supplement.pdf

**Supplement:**

[revised manuscript text omitted]

---

## Author Comment (AC1) · 26 Dec 2018

Dear referee, I wish you a Merry Christmas and a Happy New Year. Thank you very much for your valuable comments. I have tried to response to all of them base on scientific and logical reasons and hope it will meet your expectation. Best regards, Authors

Referee 1: 1) The output is a qualitative ranking of the susceptibility of the area to ground motion amplification phenomena. The Authors should stress that is a kind of

level-1 or grade-1 microzonation study according to international standards and guidelines. More detailed studies are needed in high-risk areas and should be based on numerical modelling of site effects (i.e, physically based procedures which cannot be replaced by influencing criteria methods). *R: Yes, the research is conducted based on first level seismic microzonation procedure. "The main purpose of this paper is to develop a model for evaluation of local seismic amplification using AHP, Fuzzy Logic and Weighted Linear Combination (WLC) methods in GIS. At this stage, model inputs are direct characteristics of local geology, hydrology, sedimentology, and topographical factors that should be taken into consideration. Firstly, all selected criteria were weighted using AHP method by interviewing 10 experts, next all criteria were converted into fuzzy sets, then fuzzy membership functions (MFs) were produced, finally WLC and fuzzy inference rules were applied to develop a model for producing a first level susceptibility map of local seismic amplification based on level-1 seismic microzonation procedure for a study area. "

2) My biggest doubts concern the criteria selected for the procedure. "consolidation and strength": consolidation should be replaced by stiffness (in terms of shear wave velocity or shear modulus) while the strength (i.e. soil resistance) is not pertinent; the resistance should influence the slope stability or the resistance to liquefaction not the amplification phenomena. The "particle size distribution" does not affect the amplification phenomena: the stiffness of soil is the controlling factor not the particle size: a coarse-size if loose soil can be softer than a consistent over consolidated fine grained clay ! The "depth of groundwater" is not pertinent for amplification effects, it only controls the possible occurrence of soil liquefaction in loose sandy soils.

*R: This study tries to propose a method in dealing with uncertainties. As you mentioned, there not a clear procedure of considering all influencing criteria to produce susceptibility level of local seismic amplification. Here what we are trying to say is how we can consider all criteria in a logical procedure. We know it is not a perfect way, but still give ideas to people in this field of study to work on each influencing criteria by its

own and improve the model.

The criteria were selected based on an extensive literature review. In order to be able to come up with the idea of influencing criteria on susceptibility level of local seismic amplification, we have to consider the effect of geology, geomorphology, topography and bedrock on this phenomena. We have to make separation among all influencing criteria to be able to solve our problem based on fuzzy theory. According to this theory we can handle uncertainties. As, you said regarding liquefaction and groundwater level. We are not 100 percent sure that the groundwater level will not have any effect on amplification coefficient of an area. Then, we have to come up with the idea that it may have an influence, event it is minor effect. Therefore, fuzziness can solve this problem as seen in the paper.

Yes, I understand consolidation/stiffness is the controlling factor, but still there is an effect of particle size on the local seismic amplification, which is considered through fuzziness of each criterion. Threofre, we have main controlling factors/criteria (main criteria with high weights) and minor criteria (with low weights) based on AHP method. Furthermore, sub-criteria for each criterion that are weighted based on Fuzzy Logic method, as seen in Figure. 1 spatial soft soil has fuzzy membership of 1, while hard rock has fuzzy membership (FM) of 0. It means spatial soil (unconsolidated soil and sediment) has more effect on amplification factor and areas with this type of soil will be highly amplified. Figure 2 shows the effect of grain size on amplification factor by itself (alone). Fine grain (clay) sediment has FM of 1, while coarse grain sediment (boldder) has FM of 0.3.

Let's see collateral effects on amplification factor: - Unconsolidated fine grain sediment (unconsolidated clay) = 1 * 1 = 1 - Unconsolidated boldder = 1 * 0.3 = 0.3 As seen above if we are dealing with unconsolidated clay the amplification factor will be 1, while in unconsolidated bolder the amplification factor will be 0.3.

Figure 1. Fuzzification of stiffness and strength of soil and sediment
Figure 2. Fuzzification of grain size criterion

In terms of groundwater level, as seen in Figure 3, it is the main factor in effecting liquefaction susceptibility, but it can still effect on local seismic amplification factor.

Figure 3. Fuzzification of depth of groundwater

3) In table 1 is not clear the difference between morphology of bedrock and topography of bedrock (even if, as I understand, they are considered as minor criteria). I strongly suggest to remove these factors or better define them.

*R: Morphology of bedrock defines the shape of bedrock. It can be like bowl, cylindrical, semi-spherical, conical, or even invers conical and semi-spherical as a result of revers faults.

Topography of bedrock is very similar to the topographic irregularities, it all says about roughness or smoothness areas and elevation changes.

4) Specific comments 1) It is not clear how the output of model is compared with experimental resonance frequencies. In terms of amplitude of HVSR peak with levels of shaking map ? Please describe better this validation phase.

*R: Predominant frequency ranges from 1.36 to 10.53 Hz, and amplification factor ranges between 1.33 and 4.77. These findings indicate that soil type in Bam city is mainly stiff, and amplification factor is relatively large. There is a high frequency zone from the north and northwest to the southeast of Bam where amplification factor trends to large values. However, in west and southwest of the city, fundamental frequency is low and amplification factor exhibits smaller values compared to other parts of the city.

Predominant frequency (Figure 4a and b) has been divided to 5 classes including: very low, low, moderate, high and very high. These 5 classes overlay the output of the model which has 5 classes that renege from very low to very high susceptibility levels. By comparing these two maps an error matrix will be created and overall accuracy of the output of the model will be calculated.
[Figure]

a) b) Figure 4. Control data: Dominant frequency by Lashkaripour (a) and by Motamed et al (Motamed et al., 2007) (b) using Microtremor field measurement.

5) Pag. 3: at the state of art, I don't believe that SSHA methods can be used for microzonation: source and path parameters are still quite significant to define. PSHA assessments at regional scale (at outcropping rock) and site response modelling at local (urban scale) are the most adopted procedure almost worldwide. *R: I have revised this section based on your comments as shown below: Probabilistic Seismic Hazard Analysis (PSHA) (Cornell, 1968) has been used to assess ground-motion hazards from earthquakes (Atkinson et al., 2015;Petersen et al., 2016). This method depend on "the length of the causative faults and depth of the earthquake", which are generally unknown thus causing uncertainty in assessing ground-motion of earthquakes (Wang et al., 2017). In deterministic seismic hazard analysis (DSHA) (Campbell, 2003;Atkinson and Boore, 2006) the lack of relevant ground-motion attenuation relationship for specific geographic areas can cause uncertainty in applying DSHA for assessing ground motions of an earthquake (Wang et al., 2017). Scenario-based seismic hazard analysis (SSHA) (Panza et al., 2012) applies ground-motion simulations of a scenario earthquake using specified source, path and site parameters, however the parameters needs to be defined significantly. By conducting many simulations, earthquake variability of different sources, ground-motion propagation characteristics, and local site effects can be considered. Therefore, uncertainties using SSHA are quantified explicitly (Wang et al., 2017), although this method is still under development. Therefore, PSHA assessments at regional scale and site response modelling at local scale are the most adopted procedure almost worldwide.

6) Typing 3) Pag. 11: "microtremor" instead of "microtremore *R: I have revised the word.

Please see supplement for figures.

Please also note the supplement to this comment:

https://www.nat-hazards-earth-syst-sci-discuss.net/nhess-2017-421/nhess-2017-421-AC1-supplement.pdf

---

## Author Comment (AC2) · 26 Dec 2018

Dear referee, I wish you a Merry Christmas and a Happy New Year. Thank you very much for your valuable comments. We have tried to response to all of them base on scientific and logical preseasons and hope it will meet your expectation. Best regards, Authors

Referee 2: First of all, the paper need a strong revision of the English by an English mother tongue. Second critical point, authors should avoid to repeat the same concepts

too many times just by using slightly different sentences. R: I have revised the text and removed repetition from different sections. The paper was reviewed by a mother tongue editor to make it as clear as possible.

Third, very important point. Authors talk about seismic microzonation of ground shaking amplification. This concept is not clear to me. Do they mean "seismic site response"? This is a crucial point that needs to be clarified. In addition, in the discussion section, authors suddenly introduce the concept of "susceptibility amplification" (line 469). Susceptibility is different from seismic site response! Authors need to clear state these concepts in the entire paper. R: The aim of this study was to develop a new method based on thematic layers, AHP and Fuzzy logic theory for producing a qualitative output which can rank the susceptibility of the area to ground motion amplification phenomena. I have revised the concept based on Aucelli et al. (2018) paper and applied the same terminology as they proposed " modelling of local seismic amplification susceptibility ". This concept have been used throughout of my paper.

Aucelli et al. (2018) proposed a method for producing susceptibility index to local seismic amplification in Isernia Province, Italy based on geological and geomorphological properties of studied areas. This research mostly followed an evidence based approach to estimate susceptibility level of local seismic amplification in the area, although they have not considered the use of multi-criteria decision-making methods (MCDM) in weighting and combining the influencing criteria which is the aim of conducting this research.

Fourth critical point, the method section is 12 pages long whereas the discussion and conclusion section is just 3 pages! This discrepancy is incredible! Methods section include too many repeats of the same concepts (e.g., authors said several time that they interviewed 10 experts!). R:This is right, but I had to go through a series of different steps in this section to make it as clear as possible. Each step had its own result which I could bring them to the results and discussion section of the paper, but it will make a huge separation between data preparation and fuzzification of each criterion. I had to

keep this two parts together to make a good image in readers mind.

R: I have tried to get rid of repetition through the paper. In additiond, discussion and conclusion section needs to be more detailed. R: I have read several papers and tried to discuss the findings of this research via a critical approach.

In this study, we have focused on the site effect and local geology properties of a site that have a massive influence on local seismic amplification susceptibility in the study area. To deal with related uncertainties in preparing seismic microzonation, the most important criteria were selected, weighted and then fuzzified. Criteria with high uncertainty degree such as distance of active fault to the site, depth and magnitude of the probable earthquake were not considered because there was no possibility to exactly find out where and how an earthquake will be triggered. Therefore, only the criteria with known location (x and y) and known characteristics were taken into consideration. Furthermore, to deal with uncertainties Fuzzy Logic is a suitable approach as we can define membership function of the effect of each criterion in the amplification of ground shaking by interviewing 10 experts and obtaining expert's knowledge. This can result in realistic output regarding the behavior of each criterion in ground shaking calculation. The newly developed model uses AHP and Fuzzy Logic (Zadeh, 1965) to deal with complexities and uncertainties in data analyses in weighting the criteria and fuzzifying the sub-criteria of each criterion. Although, in studies for evaluating seismic microzonation in Bangalore (India) (Sitharam and Anbazhagan, 2008), Dehli (Mohanty et al., 2007), Haldia (India) (Mohanty et al., 2007), Erbaa (Turkey) (Akin et al., 2013) and Al-Madinah (Moustafa et al., 2016) only AHP method was applied to weight the criteria, and none of these studies considered weighting of sub criteria for each criterion even using other methods. Few researchers have considered direct properties of influencing factors in assessing ground shaking amplification. Even, in evaluating seismic response developed models such as SiSeRHMap v1.0 (Grelle et al., 2016) and GIS Cubic Model (Grelle et al., 2014), the researchers have applied only lithodynamic, stratigraphic and topographic effects as influencing factors. Furthermore, Aucelli et al.

(2018) suggested a method for producing susceptibility index to local seismic ampli­fication in Isernia Province, Italy, and they have considered geological and geomor­phological properties of studied areas. Although, they have not considered the use of multi-criteria decision-making methods (MCDM) in weighting and combining the influ­encing criteria which is the aim of current study. The current research considers direct properties of each criteria and tries to manage uncertainties in criteria and sub-criteria of each criterion via weighting and fuzzification process using experts' knowledge and the use of direct properties of criteria. These processes can be extended in more details, which are subject to more investigation in the future.

Fifth point, most of the figures are very hard to read. Quality of figures should be increased. Several other comments are listed in the attached file. R: Quality and size of all figures were increased as seen below.

Please also note the supplement to this comment:
https://www.nat-hazards-earth-syst-sci-discuss.net/nhess-2017-421/nhess-2017-421-AC2-supplement.zip

---

## Referee Report (RR1)

**MAIN TEXT**

Please check references within the entire text, you often missed either a space or a semicolon to separate papers.

**Page 9:** (marine to Quaternary deposits) should be replaced by (marine to continental Quaternary deposits). Anyway, it seems to me that authors simply copied and pasted my previous comments, without clearly defining the nature of the sediment (do these sediments consist of alluvial plain gravels? Or alluvial fan gravels? Or sands? Or what else? Please clarify this point)

**Page 11:** "Geological Strength Index (Geological Survey of Iran (GSI))". It is ambiguous if GSI refers to Geological Strength Index or to Geological Survey of Iran, cite them in the correct way.

**REFERENCE LIST**

Aucelli et al. (2018) is listed in the text but it is missing in the reference list

Fah et al. (1997) is listed twice in the reference list

"Report on the Bam earthquake: http://www.ncc.org.ir/homepage.aspx?site=NCCPortal&tabid=1&lang=fa-702 IR, access: May 15, 2013, 2003." is listed in the references but not in the text. In addition, I checked the link and it does not contain a direct and immediate connection to this Report. So, I suggest to remove from the reference list or, in alternative, to cite it correctly to provide a direct link to the Report.

Please check the reference list and list references in the correct form, according to NHEES guidelines (e.g., Fraume et al., 2014; Rehman et al., 2016; Teramo et al., 2005).

---

## Author Response (AR2)

**Response to reviewers' comments:**

Dear referees,

We greatly appreciate your valuable comments on our manuscript and suggestion of very good papers. We have tried to address each of your comments in the revised manuscript base on reviewing suggested papers and scientific reasons, and hope the revised version of the manuscript will meet your expectation.

Our responses to your comments are listed below in italics following each specific comment.

Best regards,

Authors

**Referee 1**

1- Please check references within the entire text, you often missed either a space or a semicolon to separate papers.
*\* References were revised and also double checked within the text and reference list.*

**2- Page 9:** (marine to Quaternary deposits) should be replaced by (marine to continental Quaternary deposits). Anyway, it seems to me that authors simply copied and pasted my previous comments, without clearly defining the nature of the sediment (do these sediments consist of alluvial plain gravels? Or alluvial fan gravels? Or sands? Or what else? Please clarify this point)
*\* This has been revised as seen below:*
*\* In the northern part of the city, the sediment (marine to continental Quaternary deposits including alluvial plain gravels with interlayered clay, silt and sand) thickness ranges from 0 m, where bedrock is exposed beneath Arg-e-Bam, to 90 m across most of the northern half of the study area.*

**3- Page 11:** "Geological Strength Index (Geological Survey of Iran (GSI))". It is ambiguous if GSI refers to Geological Strength Index or to Geological Survey of Iran, cite them in the correct way.
*\* It has been revised as Geological Strength Index (GSI), and GSI only refers to Geological Strength Index.*

4- Aucelli et al. (2018) is listed in the text but it is missing in the reference list.
*\* Ii has been added to the reference list.*
*Aucelli, P. P. C., Di Paola, G., Valente, E., Amato, V., Bracone, V., Cesarano, M., Di Capua, G., Scorpio, V., Capalbo, A., and Pappone, G.: First assessment of the local seismic amplification susceptibility of the Isernia Province (Molise Region, Southern Italy) by the integration of geological and geomorphological studies related to the first level seismic microzonation project, Environmental earth sciences, 77, 118, 2018.*

5- Fah et al. (1997) is listed twice in the reference list.
*\* It has been revised.*
*\* Fäh, D., Rüttener, E., Noack, T., and Kruspan, P.: Microzonation of the city of Basel, Journal of Seismology, 1, 87-102, 10.1023/a:1009774423900, 1997.*

6- "Report on the Bam earthquake:
http://www.ncc.org.ir/homepage.aspx?site=NCCPortal&tabid=1&lang=fa-
702 IR, access: May 15, 2013, 2003." is listed in the references but not in the text. In addition, I checked the link and it does not contain a direct and immediate connection to this Report. So, I suggest to remove from the reference list or, in alternative, to cite it correctly to provide a direct link to the Report.
*\* It has been replaced by another reference.*
*\* National Cartographic Center of Iran (NCCI): Report on the Bam earthquake, Tehran, 98, 2003*

7- Please check the reference list and list references in the correct form, according to NHEES guidelines (e.g., Fraume et al., 2014; Rehman et al., 2016; Teramo et al., 2005).
*All references were doubled checked and revised based on NHEES guidelines.*

**Referee 2**

1- The paper can be published after some corrections.
2- Page 2: Expand the part related to seismic microzonation studies considering the following paper:

*The suggested references were reviewed and added in the text.*

MERM microzonation manual (2003) sets different criteria effecting the amplitude and duration of ground shaking at a specific site. These include "the magnitude of the earthquake, focal point and depth of the earthquake, directivity of the energy released, distance of rupture from the site, geological condition from the site to the location of the earthquake, local geological settings, geotechnical properties, and topographical condition of the site" (SM Working Group, 2015;Boore, 2003;Hassanzadeh et al., 2013;Castelli et al., 2016a;Castelli et al., 2016b).

In this research, Analytical Hierarchal Process (AHP) (Saaty, 1980) has been utilize as it is one of the most useful method in calculating criteria's weights, and AHP in combination with GIS were applied to produce seismic microzonation map of Bangalore (Sitharam and Anbazhagan, 2008) (2008), Dehli (Mohanty et al., 2007), Haldia, Bengal Basin (India) (Mohanty and Walling, 2008), Erbaa (Turkey) (Akin et al., 2013) and Al-Madinah (Moustafa et al., 2016) and generating ground-shaking map for Catania (Italy) using GIS (Castelli et al., 2016a).

3- Page 2: In the list of earthquakes, insert also the one of L'Aquila considering the following paper:

*The suggested references were reviewed and added in the text.*

It has long been known that local conditions of foundation soils have a significant impact on the effects of an earthquake on building destruction level, as it was demonstrated in previous earthquakes such as Mexico City, 1985 (Beck and Hall, 1986), Kobe, 1995 (Wald, 1996), Izmit, 1999 (Tang, 2000), Umbria-Marche earthquake, 1997 (Moro et al., 2007) and Bam earthquake, 2003 (Ramazi and Jigheh, 2006) and L'Aquila earthquake, 2009 (Monaco et al., 2012;Capilleri et al., 2014) and buildings that were located on unconsolidated sediments had greater destruction levels (Ramazi and Jigheh, 2006).

4- Page 3: "behid" is perhaps "behind".

*It has been revised.*

* These are motivations behind conducting this research.

5- Page 9: "descried" is perhaps "described".

* It has been revised. To conduct this analysis, 10 experts were interviewed regarding membership degree of sub criteria of each criterion, and mode of each sub criteria was calculated and MFs for each criterion was depicted as described in the following:

6- Page 7 and 18: The reference "Aucelli et al., 2018" is absent in the References.

*I have added Aucelli et al. (2018) to the reference list.*
*Aucelli, P. P. C., Di Paola, G., Valente, E., Amato, V., Bracone, V., Cesarano, M., Di Capua, G., Scorpio, V., Capalbo, A., and Pappone, G.: First assessment of the local seismic amplification susceptibility of the Isernia Province (Molise Region, Southern Italy) by the integration of geological and geomorphological studies related to the first level seismic microzonation project, Environmental earth sciences, 77, 118, 2018.*

7- Page 11: Expand the part related to liquefaction considering the following paper:

*The suggested references were reviewed and added in the text.*

Research on the effects of groundwater shows it can magnify an earthquake's damage. The most well known effect is liquefaction. The geologic and hydrologic factors that affect liquefaction susceptibility are the age and the type of sedimentary deposits, the looseness of cohesions and the depth to the groundwater table (Tinsley et al., 1985;Cavallaro et al., 2018).

8- Page 11: "Geological Survey of Iran (GSI)" is better "Geological Survey of Iran (GSI), 1993".

*It is revised as Geological Strength Index (GSI), and GSI only refers to Geological Strength Index.*

9- Page 12: Expand the part related to 2d numerical analysis considering the following paper:

*The suggested references were reviewed and added in the text.*

* The effects of slope angle on topographic amplification factor was investigated by Bisch et al. (2012), and they classified the slope angle into three categories with different effect level including: 0-15 with no effect,  15-30 degree with 1.2  (coefficient) and more than 30 degree with 1.4 amplification coefficient. Although, Cavallaro et al. (2012) suggested that topographic amplification factor can be considerable for slope even less than 15 degree.

* In addition, several calibration constants should be calculated using 2d numerical analysis for each study area to compute topographic effects on local seismic amplification. Cavallaro et al. (2008) investigated 2d model for analysing site response of the Monte Po Hill in the City of Catania considering the effect of topographic and stratigraphic properties on the amplification factor in an area. They concluded that near to the slop crest, the effect of topographic properties on amplification factor is more relevant than stratigraphic property.

10- Page 14 and 17: The reference "National Cartographic Center (NCC), 2003" is absent in the References.

*It has been revised.*
*National Cartographic Center of Iran (NCCI): Report on the Bam earthquake, Tehran, 98, 2003*

11- The reference "Fath et al., 1997" is repeated twice.

*It has been revised.*
*Fäh, D., Rüttener, E., Noack, T., and Kruspan, P.: Microzonation of the city of Basel, Journal of Seismology, 1, 87-102, 10.1023/a:1009774423900, 1997.*

---

## Author Response (AR3)

**Response to editor**

Dear Dr. Oded Katz,

We greatly appreciate your valuable comment on our manuscript. We have included all figures in a zip folder with high quality and readable legends.  We hope the revised version of the manuscript will meet your expectation.

Best regards,

Authors

Comment:

Make sure the legend of Figures 7 - 14 is in higher resolution and thus readable.

* We have included all figures in a zip folder with high quality and readable legends.

.